# A Comparative Analysis of Morphology and Dimensions of Functional Blebs following PRESERFLO-Microshunt and XEN-Gel-Stent, a Study Using Anterior Segment OCT

**DOI:** 10.3390/diagnostics13142318

**Published:** 2023-07-09

**Authors:** Somar M. Hasan, Theresa Theilig, Menelaos Papadimitriou, Daniel Meller

**Affiliations:** Department of Ophthalmology, Jena University Hospital, 07747 Jena, Germanydaniel.meller@med.uni-jena.de (D.M.)

**Keywords:** glaucoma filtering surgery, minimally invasive glaucoma surgery, bleb surgery, optical coherence tomography, bleb morphology, PRESERFLO Microshunt, XEN Gel Stent

## Abstract

Evaluation of bleb morphology is a vital part of successful filtration glaucoma surgery. The PRESERFLO-MicroShunt (PRESERFLO) and XEN-Gel-Stent (XEN) are drainage devices implanted using different surgical approaches (ab externo and ab interno, respectively), potentially resulting in distinct bleb morphology. Understanding these morphological differences is essential for postoperative care. In this study, we retrospectively examined functioning blebs following PRESERFLO and XEN implantation using high-resolution anterior segment OCT imaging. Qualitative assessment utilizing the Jenaer Bleb Grading System and quantitative assessment measuring 12 parameters representing bleb dimensions were conducted, and the results were compared between the two groups. A total of 80 eyes from 80 patients were included (41 after PRESERFLO, 39 after XEN). Functioning blebs following PRESERFLO exhibited a higher frequency of hyperreflective tenon changes compared to XEN (31.7% vs. 10.3%, respectively, *p* = 0.02) and a lower frequency of cavernous changes (17.1% vs. 35.9%, *p* = 0.05). Additionally, PRESERFLO blebs showed a higher frequency of visible episcleral lakes (92.7% vs. 30.8%, *p* < 0.001). Furthermore, PRESERFLO blebs demonstrated larger height (2.13 ± 0.5 vs. 1.85 ± 0.6 mm, *p* = 0.03), width (10.31 ± 2.3 vs. 9.1 ± 2.3 mm, *p* = 0.02), length (9.13 ± 1.8 vs. 8.24 ± 1.9 mm, *p* = 0.04), posterior location relative to the limbus (6.21 ± 1.2 vs. 5.21 ± 1.8 mm, *p* = 0.005), and a thicker bleb wall (1.60 ± 0.5 vs. 1.1 ± 0.4 mm, *p* = 0.004). Functioning blebs following PRESERFLO and XEN displayed morphological distinctions, likely attributed to variations in surgical techniques (ab externo vs. ab interno) and stent dimensions. These morphological differences should be taken into consideration when evaluating blebs, as they could impact assessments of bleb functionality and influence decisions regarding postoperative interventions.

## 1. Introduction

The assessment of bleb morphology is a crucial aspect of postoperative care in filtration surgery, as it guides the modification of the wound healing process and aims to improve surgical outcomes [1]. Various grading systems have been introduced to morphologically classify blebs that occur after trabeculectomy (TE) [2,3,4]. Traditionally, these systems have relied on biomicroscopy and photography. However, the advent of anterior segment optical coherence tomography (AS-OCT), particularly high-resolution systems, has provided a new tool for examining blebs, revealing ultrastructural changes and enabling the objective documentation of their morphology. Several authors have utilized AS-OCT to develop classification systems and compared them to biomicroscopy-based systems, showing good correlations [5,6], as well as prognostic value for bleb function [7,8].

The introduction of minimally invasive glaucoma surgery (MIGS) and minimally invasive bleb surgery (MIBS) has revolutionized the surgical management of glaucoma. Through MIBS, implanted stents offer a standardized and simplified surgical technique for draining aqueous humor from the anterior chamber into the subconjunctival space, in contrast to traditional trabeculectomy (TE). Two commonly used MIBS procedures are the PRESERFLO™-Microshunt (PRESERFLO, Santen, Osaka, Japan), implanted via an ab externo approach, and the XEN^®^45-Gel-Stent (XEN, Allergan, an AbbVie company, North Chicago, IL, USA), primarily implanted via an ab interno approach. Both stents result in a bleb whose evaluation is just as important as that after TE. However, due to the differing surgical techniques and stent dimensions, varying drainage capabilities and characteristics of the resulting blebs are expected. Despite this, limited knowledge exists regarding the differences in bleb morphology following the implantation of these stents. Studying these variations is of clinical significance, as morphological differences may impact the decision-making process in the postoperative phase, ultimately influencing surgical success.

## 2. Materials and Methods

This is a retrospective comparative study that included eyes of patients with uncontrolled glaucoma despite maximal tolerated medications. These patients underwent one of two procedures as a stand-alone surgery between September 2018 and September 2022, and were divided into two groups: the PRESERFLO-Group, which involved the implantation of PRESERFLO with Mitomycin-C (MMC) using an ab externo approach, and the XEN-Group, which involved the implantation of XEN with MMC using an ab interno approach. The decision of which procedure to perform on each patient was primarily based on the availability of the stents, with XEN being more commonly available from September 2018 to July 2020, and PRESERFLO being the predominant option thereafter.

Functioning blebs of patients attending follow-up visits from the 5th week post surgery onwards were examined using anterior segment optical coherence tomography (AS-OCT). Blebs during the first 4 weeks post surgery were excluded due to the rapid morphological changes that occur during this early phase [9,10]. A functioning bleb was defined as having an intraocular pressure (IOP) of ≤21 mmHg and a reduction of ≥20% from the pre-operative IOP (at the time of surgical indication), without an increase in the number of glaucoma medications (NoM). The study excluded dysfunctional blebs due to their characteristics: either flat blebs with no observable measurable anatomical changes (making comparison impossible), or encapsulated blebs where tenon cysts were visible, but exhibited similar morphology regardless of the stent or surgical technique employed. 

Eyes that had undergone prior ocular surgery affecting the conjunctiva (such as prior glaucoma surgery, strabismus surgery, vitrectomy, or pterygium excision), as well as eyes that required bleb revision or needling due to bleb failure, were also excluded. In cases where both eyes were eligible for the study, the eye that underwent surgery first was selected.

Collected preoperative data included age, sex, glaucoma type, glaucoma localization, IOP at the time of surgical indication, and the number of glaucoma medications (NoM). Postoperative data included the date of surgery, date of examination, IOP, NoM, and any postoperative complications. Each bleb was qualitatively and quantitatively examined and the differences between the two groups were analyzed.

### 2.1. Surgical Technique

Preoperative preparation: All glaucoma medications are discontinued for 4 weeks prior to surgery, and patients are prescribed dexamethasone eyedrops (Dexapos COMOD 1.0 mg/mL eye drops, Ursapharm, Saarbrücken, Germany) three times daily, along with oral Acetazolamide 250 mg (Glauopax^®^ 250 mg tablets, Omni-Vision, Puchheim, Germany). The dosage of Acetazolamide is adjusted based on intraocular pressure (IOP) measurements.

Implantation of XEN45-Gel-Stent with the application of MMC: After disinfection, the conjunctiva is marked 3 mm from the limbus. A subconjunctival injection of 0.1 mL of MMC (0.2 mg/mL) is administered approximately 5 mm posterior to the limbus. The resulting cyst is gently massaged both posteriorly and anteriorly, ensuring not to reach the limbus. A 2.0 mm inferotemporal clear corneal incision is made, along with a nasal paracentesis. The anterior chamber is filled with cohesive viscoelastic, and the XEN45-Gel-Stent is inserted anterior to the trabecular meshwork into the subconjunctival space, aiming for the 3 mm mark on the conjunctiva. Intraoperatively, the location and distance of the stent from the cornea and iris are assessed using gonioscopy. The distal segment of the stent is examined for mobility using a spatula. With this approach, the subconjunctival placement of the stent is typically achieved in most cases. However, due to the conjunctiva not being opened, the exact localization of the outer segment is not possible.

Implantation of PRESERFLO-MicroShunt with the application of MMC: After disinfection, a fixation suture (7-0 Vicryl, Ethicon, Somerville, NJ, USA) is inserted into the cornea at 12 o’clock, and the surgical field is exposed. A peritomy is performed superiorly over 2 clock hours with 2 radial cuts in the conjunctiva. Tenon tissue is then carefully dissected horizontally and posteriorly from the sclera, and the episcleral vessels are cautiously cauterized. Two Lasik cornea shields are soaked in MMC solution (0.2 mg/mL) and placed under the tenon tissue for 3 min. After the irrigation of the surgical field with 20 mL of balanced salt solution, the sclera is marked 3 mm posterior to the limbus, and a scleral tunnel of approximately 2 mm in length is created using the included knife. The 25-gauge needle is inserted into the anterior chamber through the tunnel, ensuring proper distance from the cornea and iris. The PRESERFLO-MicroShunt is then inserted into the anterior chamber through the tunnel. After confirming proper drainage, the tenon tissue is repositioned anteriorly and secured to the sclera with two interrupted sutures (10-0 Vicryl), followed by conjunctival closure with two to four interrupted sutures (10-0 Vicryl).

Postoperative regimen: Ofloxacin eye drops (Floxal^®^ 3 mg/mL eye drops, Bausch&Lomb, Laval, QC, Canada) are administered five times daily for one month, and dexamethasone eyedrops (Dexapos COMOD 1.0 mg/mL eye drops, Ursapharm, Saarbrücken, Germany) are given every two hours during the first week, followed by five times daily starting from day 8. The dosage of dexamethasone is gradually reduced by one drop per day every four weeks. Accordingly, patients receive dexamethasone eye drops for a total of five months postoperatively.

### 2.2. Bleb Examination Using AS-OCT

A high-resolution swept-source AS-OCT (ANTERION^®^, Heidelberg Engineering GmbH, Heidelberg, Germany) was used to examine the blebs following a standardized protocol. The patient was instructed to look downwards and to either temporally or nasally expose the bleb. The upper lid was gently lifted using a cotton swab, taking care not to apply any pressure on the eye. A rectangular box, 7.5 mm wide, from the imaging module with 45 scans was positioned at the center of the stent. Active eye tracking was turned off during the scans. Two sets of scans were performed: the first set with scans oriented parallel to the stent (mostly radial to the limbus), and the second set with scans oriented perpendicular to the stent (mostly tangential to the limbus), as shown in Figure 1. Care was taken to capture the maximum visible area of the bleb posteriorly and horizontally. This resulted in 90 scans for each bleb, which were evaluated by the same examiner (SMH) who was blinded to the clinical data.

Descriptive assessment: Images were qualitatively classified according to the Jenaer Bleb Grading System (JBGS) [11]. This grading system allows for the documentation of morphological changes at three anatomical levels: the conjunctiva (C0 = no changes, C1 = intraepithelial cysts, and C2 = subconjunctival spaces), tenon’s tissue (T0 = no changes, T1 = hyperreflective changes, T2 = hyporeflective changes, and T3 = cavernous changes), and the episcleral space (ES0 = no episcleral space and ES1 = episcleral space visible). The classification is based on comparing the changes observed in each bleb to standard images and assigning a corresponding code for each layer. For example, a bleb with subepithelial spaces (C2), hyporeflective changes in tenon’s tissue (T2), and a visible episcleral space (ES1) would be classified as C2T2ES1.

Quantitative assessment: Images were exported and analyzed using photo editing software (Adobe Photoshop CC, Version 20.0.0, Adobe Systems Incorporated, San Jose, CA, USA). The width of the image, as provided by the AS-OCT machine, was entered into the scaling tool of the editing software. Twelve parameters were measured for each bleb when applicable, as described in Table 1 and corresponding Figure 1. These parameters were selected to assess the dimensions of the bleb in three geometric directions (height, width, and length). This included measurements of the overall bleb size (MBH, MBW, and MBL), as well as measurements of intra-bleb components such as the size of the episcleral space (MLH, MLW, and MLL) or the cavity in the case of a cavernous form (MCH, MCW, and MCL). The thickness of the bleb wall, a well-established parameter of bleb function (BWT-L and BWT-C), and the location of the bleb (distance to the limbus, DtL) were also measured. Each of the 90 images of each bleb was evaluated for the corresponding parameters.

Data were analyzed using IBM^®^ SPSS^®^ Statistics (Version 22.0, IBM Corp., Armonk, NY, USA). All numerical parameters were studied for normal distribution. Descriptive and quantitative parameters were compared between the PRESERFLO-Group and the XEN-Group using the Pearson Chi-Square, independent t-test or the Mann–Whitney U test, as applicable. Comparison of descriptive parameters was performed for each pattern separately (e.g., presence of subconjunctival spaces, pattern C2,to its absence). A *p*-value ≤ 0.05 was considered significant.

Informed consent was obtained from subjects included. The study was conducted in compliance with the tenets of the Declaration of Helsinki and approved by the local ethical committee of the Jena university hospital, Reg.-Nr. 2023-3018-Daten.

## 3. Results

Included were 80 eyes of 80 patients (41 in the PRESERFLO-Group and 39 in the XEN-Group). Patients’ characteristics and demographic data can be found in Table 2.

### 3.1. Results of Descriptive Assessment

When comparing the two groups in terms of qualitative changes (Table 3), there were no significant differences between the groups in the frequency of conjunctival changes (*p* > 0.05 for all comparisons, presence of a conjunctival pattern compared to its absence). At the tenon level, the most common pattern observed in both groups was hyporeflective changes (T2), with a prevalence of 51.2% in the PRESERFLO-Group and 51.3% in the XEN-Group (*p* = 1.0). The PRESERFLO-Group had a higher frequency of hyperreflective tenon changes (T1, *p* = 0.02) and a lower frequency of cavernous changes (T3, *p* = 0.05) compared to the XEN-Group, while there was no difference in the frequency of the T0 pattern. The presence of an episcleral lake was also more frequent in the PRESERFLO-Group (ES1, *p* < 0.001). The most common bleb form in the PRESERFLO-Group was C2T2ES1, accounting for 34.1% of all blebs, followed by C2T3ES1 with 14.6%. In the XEN-Group, the most common form was C2T2ES0, comprising 25.6% of all blebs, followed by C2T3ES0 with 17.9% (Figure 2).

### 3.2. Results of Quantitative Assessment

Comparing the quantitative parameters between both groups showed that the MBH, MBW and MBL were larger in the PRESERFLO-Group compared to the XEN-Group (*p*= 0.027, 0.017 and 0.04, respectively). The MLL was larger in the PRESERFLO-Group compared with the XEN-Group while the MLH and MLW did not differ significantly (*p* = 0.014, 0.11 and 0.25, respectively). The MCW was larger in the PRESERFLO-Group (*p* = 0.038), while the MCH and the MCL were comparable with those of the XEN-Group. The BWT-L and the BWT-C were significantly higher in the PRESERFLO-Group compared with the XEN-Group (*p* = 0.004, 0.001, respectively). The DtL was also larger in the PRESERFLO-Group compared to the XEN-Group (*p* = 0.005). Results of the quantitative assessment are listed in Table 4.

## 4. Discussion

In this study, we conducted a comparison of the morphology of functioning blebs following PRESERFLO and XEN stents using both descriptive and quantitative approaches. The two groups were similar in terms of preoperative parameters; however, after surgery, the PRESERFLO-Group exhibited significantly lower intraocular pressure (IOP) and a lower number of glaucoma medications (NoM) compared to the XEN-Group. Similar findings have been reported in previous studies when comparing PRESERFLO with XEN, particularly in the initial months post surgery [10,12]. However, it is worth noting that these differences tend to diminish over time. The disparity in our study compared to others could be attributed to the exclusion of eyes with non-functioning blebs and those that underwent needling or bleb revision, which were included in other studies.

For the descriptive comparison, we utilized the Jenaer Bleb Grading System (JBGS), which we introduced earlier, to qualitatively evaluate the morphology of blebs using AS-OCT. This grading system enables the comprehensive documentation of changes in all layers of the bleb, allowing for a more precise description compared to other AS-OCT-based grading systems. Many existing systems primarily focus on major changes while disregarding other morphologies [13,14]. By applying the JBGS, we found that the frequency of conjunctival changes in functioning blebs following PRESERFLO and XEN stents were comparable. The majority of functioning blebs exhibited subconjunctival spaces (C2), followed by intraepithelial cysts (C1). Traditionally, the presence of conjunctival cysts has been considered a positive indication of bleb function [4] However, their assessment has relied on biomicroscopy, which is likely to detect only larger cysts. Using AS-OCT, we observed that subconjunctival spaces (subconjunctival separation) were most commonly seen in the early postoperative phase after both XEN and PRESERFLO [9,10]. However, their frequency decreased over time.

In our cohort of functioning blebs, the majority of eyes exhibited subepithelial spaces, which has been correlated with reduced intraocular pressure (IOP) in other studies [9]. However, in one of our previous studies, we were unable to establish a correlation between conjunctival changes observed using AS-OCT and postoperative IOP in blebs following XEN surgery [11]. Interestingly, the surgical approach (ab externo versus ab interno) does not appear to influence the frequency of conjunctival changes in functioning blebs.

At the tenon level, the most prevalent pattern in both groups was hyporeflective changes (T2), which accounted for over 50% of the blebs in each group; this comes in accordance with published data, showing a significant correlation of hyporeflective changes and functional blebs [15]. In the PRESERFLO-Group, the frequency of hyperreflective changes (T1) was nearly three times higher compared to the XEN-Group, while the frequency of cavernous changes (T3) was significantly lower. The rates of hyporeflective changes and the absence of tenon changes were comparable between the two groups.

Regarding the episcleral level, we observed the presence of an episcleral lake (ES1) in the PRESERFLO-Group three times more frequently than in the XEN-Group. Only 7.3% of eyes with a functioning bleb after PRESERFLO showed no episcleral lake, in contrast to 69.2% in the XEN-Group.

These changes at the tenon and episcleral level can most likely be explained through the different surgical approaches used: increased frequency of higher reflectivity, decreased frequency of cavernous changes along with a higher rate of an existent episcleral lake in the PRESEFRLO-Group compared with XEN-Group is most likely a result of the ab externo approach where a wide dissection of the tenon capsule from the sclera was performed. The application of MMC in this surgically created cavity (the episcleral space) seems to direct the drainage into this space, making it the main container of drained aqueous fluid and results in less fluid accumulation inside tenon layers and thus also results in more hyperreflectivity of the tenon capsule. The lack of surgical manipulation at the bleb area during the ab interno approach results in the tenon capsule being the main reservoir of the drained aqueous fluid. This manifests with an increased frequency of hyporeflectivity (right down to higher rates of cavernous changes) in the tenon layers along with a low frequency of a visible episcleral lake. The hyperreflectivity per se was not an indicator of a reduced bleb function in the PRESERFLO-Group, an important difference to what is known after XEN or TE, where hyperreflective changes of the bleb wall were correlated with a worse bleb function [16,17]. Still, most studies dealing with bleb morphology using AS-OCT focus on the bleb wall reflectivity (uniform versus multiform) as an indicator of bleb function without always taking the episcleral lake into consideration [9,10,13,18]. This might be misleading in case of blebs following PRESERFLO. Dangda et al. [19] observed the presence of an episcleral lake in all functioning blebs following the implantation of XEN through an ab externo approach and we found similar results in functioning blebs after open conjunctival revision (which resembles an ab externo approach) in failed XEN cases [20] as we found a significantly higher frequency of hyperreflective tenon changes (T1) and visible episcleral lakes (ES1).

Comparing the quantitative parameters of both groups showed that functioning blebs in the PRESERFLO-Group were higher (larger MBH), wider (MBW) and longer (MBL) than those in the XEN-Group. The episcleral lake (when present) was significantly longer (MLL) in the PRESERFLO-Group with comparable height and width (MLH and MLW). A cavity seen in the cavernous form of the bleb was significantly wider (MCW) in the PRESERFLO-Group with comparable height and length (MCH and MCL). 

Very little is known about the dimensional differences of blebs after PRESERFLO and XEN. The horizontal (width) and vertical (height) diameters seem to increase in the early postoperative phase after both stents [10,14]. Gambini et al. [10] found, in a comparison study 6 months post operation, that “vertical diameter” (corresponding to MBH in our study) was larger following PRESERFLO compared to XEN. This comes in accordance with our results. Miguel A. Teus et al. also observed similar results when comparing blebs after XEN and TE [15]. The “horizontal diameter” (corresponding to MBW) was, however, larger following XEN, which contradicts with our results. Nevertheless, the definition of horizontal diameter was different to that in our study as Gambini et al. used microcysts, layers of aqueous flow and a posterior aqueous lake to measure the horizontal diameter. We measured the whole width of the bleb between the points where tenon changes started and ended. In addition to that, the measurements in the Gambini et al. study were performed on two images where better quality and visibility of the bleb were observed [10]. We examined all 90 images of each bleb and searched for the maximal or minimal measurements of each of the 12 parameters in each scan. 

Different bleb dimensions are most likely a result of the surgical technique, as the wide dissection of the episcleral space and the application of MMC in this space is most likely to result in reduced fibrosis and delayed wound healing, which reduces the resistance of tissue and results in a longer (more extension posteriorly) episcleral lake. However, comparable height and width of the episcleral lake suggests no changes of resistance at these axes. This observation is supported by similar findings in an ex vivo study by Lee RMH et al. [21] as they found the reduced resistance of the subconjunctival space in the ab externo approach compared to the ab interno approach. It is noteworthy that the PRESERFLO group exhibited lower intraocular pressure (IOP) when compared to the XEN group. The decrease in IOP could be either the cause or the consequence of the larger bleb dimensions observed after PRESERFLO. An alternative explanation could be attributed to the ab externo approach employed, which might have led to reduced resistance in the episcleral and/or intratenon space. This could potentially account for both the lower IOP and the larger bleb dimensions simultaneously.

The thickness of the bleb wall is a well-established parameter of bleb function after trabeculectomy [7,16,22], as a thicker wall correlated with lower IOP in different studies. In our cohort of functioning blebs, we found that the wall thickness in the PRESERFLO group was significantly higher than that of the XEN eyes, whether measured from the episcleral lake [16] or from the cavity of blebs with cavernous form. Regarding the localization of the bleb, blebs of the PRESERFLO group were more posteriorly located to limbus (highest point of the bleb) compared with those of the XEN group. 

Factors that might have contributed to different bleb wall thicknesses and bleb locations are the design and dimensions of the stents. The PRESERFLO stent is longer than the XEN stent (8.5 versus 6 mm) and is larger in outer (350 versus 150 µm) and inner (70 versus 45 µm) diameter. This leads to different drainage potentials which might have affected the tenon morphology as the tissue absorbing the aqueous fluid [21]. The distal orifice of the PRESERFLO is located more posterior to limbus compared with XEN (about 6 mm posterior to limbus of PRESERFLO compared to 5 mm for XEN). This might have affected the localization of the bleb. The more posterior location of the bleb where the tenon capsule is thicker along with surgically securing the PRESERFLO deep in the subtenon space might have resulted in increased wall thickness as the whole tenon is covering the stent in the case of PRESERFLO. In case of XEN, we deal with a variable depth of stent implantation in the subconjunctival space because of the ab interno approach, which along with the more anterior stent location (thinner tenon), might have resulted in thinner bleb wall. 

Our study has some limitations. It has a retrospective design. A confirmation of the results using a prospective study is recommended. We also examined each bleb on a single time-point. Many authors describe a changing bleb morphology over time. However, changes are more dynamic in the early postoperative phase [9,10] which we excluded in our study. We also excluded blebs which have undergone needling or revision, so that our results might not be applicable in these cases. The manual measurement of bleb parameters along with descriptive classification can be influenced by subjective factors and lead to bias. Still, we tried to clearly define each parameter measured and used standard images for descriptive classification to reduce bias as much as possible. However, this study is the first one to our knowledge which compares the morphology of functioning blebs following PRESERFLO and XEN using a standardized descriptive and quantitative assessment of bleb and delivers clinically relevant information for the practicing ophthalmologist in the postoperative phase following implantation of these stents. 

## Figures and Tables

**Figure 1 diagnostics-13-02318-f001:**
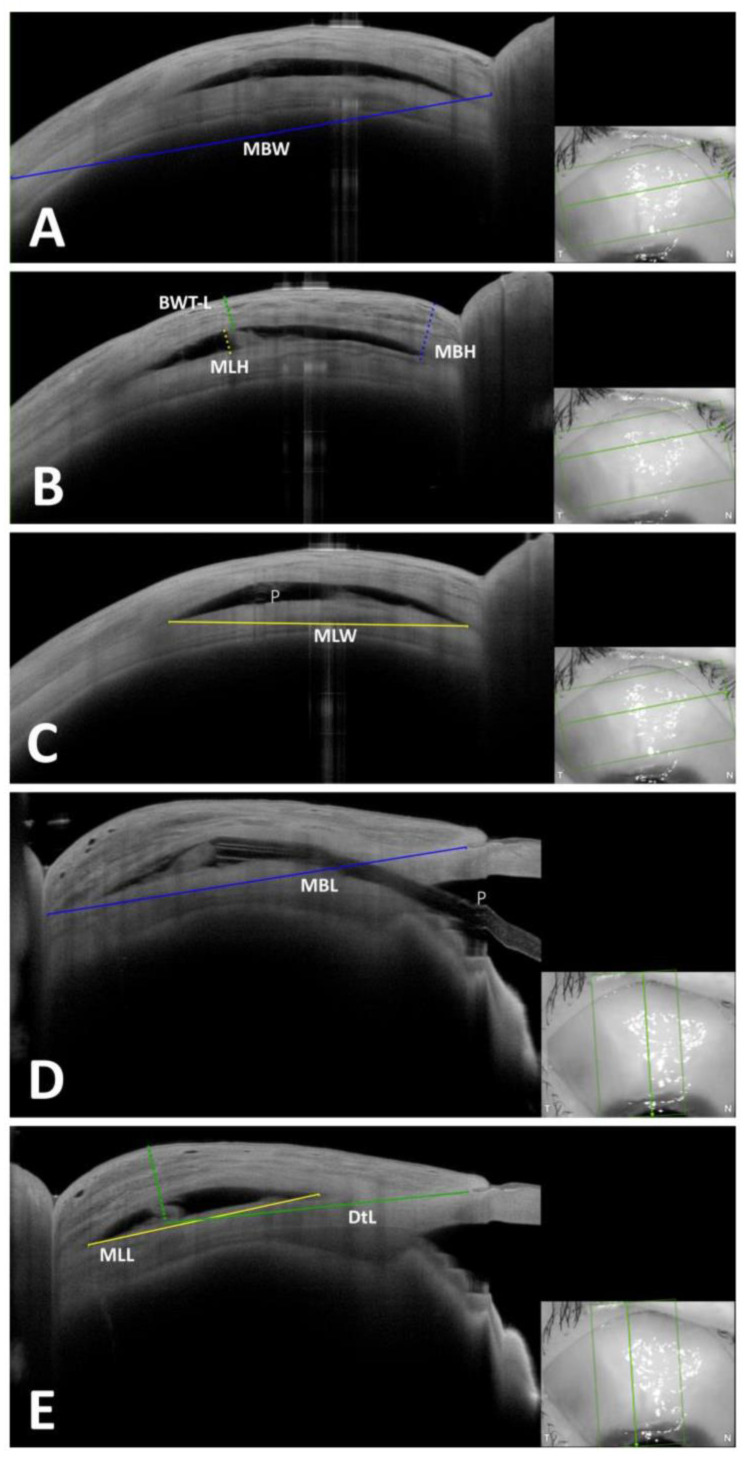
Example of measured quantitative parameters in a bleb after PRESERFLO. insets: scanned areas shown inside the green rectangles. (**A**) Tangential scan where MBW was measured (blue continuous line). (**B**) Tangential scan where MBH (blue dotted line), MLH (yellow dotted line) and BWT-L (green dotted line) were measured. (**C**) Tangential scan where MLW (yellow continuous line) was measured. (**D**) Radial scan where MBL (continuous blue line) was measured. (**E**) Radial scan where MLL (yellow continuous line) and DtL (green continuous line) were measured. P: PRESERFLO.

**Figure 2 diagnostics-13-02318-f002:**
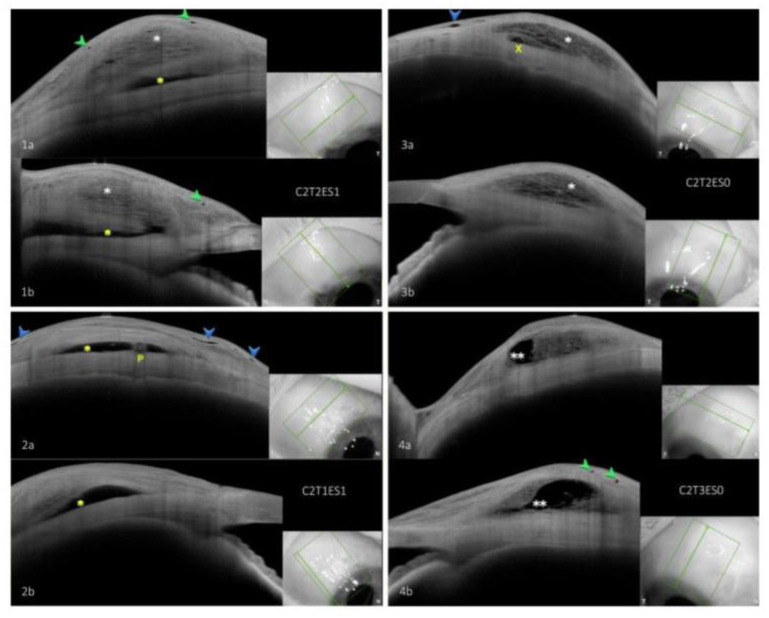
Examples of most common bleb forms in each group. insets: scanned areas shown inside the green rectangles. (**1a**,**1b**) C2T2ES1, tangential and radial scans of most common bleb form following PRESERFLO showing conjunctival cysts, however, subconjunctival spaces were seen in other scans of this bleb. (**2a**,**2b**) C2T1ES1, tangential and radial scans showing second most common bleb form after PRESERFLO. (**3a**,**3b**) C2T2ES0, tangential and radial scans of most common bleb form following XEN. (**4a**,**4b**) C2T3ES0, tangential and radial scans of second most common bleb form following XEN. Green arrowhead: intraepithelial cysts, blue arrowhead: subconjunctival spaces, white asterisk: hyporeflective tenon changes and episcleral lake, yellow asterisk: hyperreflective tenon changes, double white asterisk: cavernous tenon changes, P: PRESERFLO-MicroShunt, X: XEN-Gel-Stent.

**Table 1 diagnostics-13-02318-t001:** Definitions of quantitative parameters measured in this study.

Parameter	Abbreviation	Description	Remarks
**Maximum Bleb Height**	MBH	The maximum height of the bleb seen in the tangential scans, measured as the maximum perpendicular distance from the sclera to the first reflex at the conjunctiva.	
**Maximum Bleb Width**	MBW	The maximum width of the bleb seen in tangential scans, measured as a direct line between two points: the beginning of changes in tenon thickness nasally to the end of the tenon changes temporally.	If the whole width of the bleb could not be captured in a single image, the maximum visible width was measured.
**Maximal Bleb Length**	MBL	The maximum posterior extension of the bleb seen in radial scans, measured as a direct line between two points: from the first changes in tenon thickness anteriorly to the last visible tenon change posteriorly. If the bleb extended over the cornea, measurement was started at the level of the scleral spur.	If the whole length of the bleb could not be captured in a single image, the maximum visible length was measured.
**Maximum Lake Height**	MLH	The maximum height of the episcleral lake (ES1 according to JBGS) in the tangential scans, measured as the maximum perpendicular distance from the inferior to the superior edge of the episcleral lake.	MLH was measurable only in blebs showing the pattern ES1
**Maximum Lake Width**	MLW	The maximum width of the episcleral lake seen in tangential scans, measured as a direct line between two points: the beginning of the episcleral lake nasally to its end temporally.	MLW was measurable only in blebs showing the ES1-Pattern
**Maximal Lake Length**	MLL	The maximum posterior extension of the episcleral lake seen in a radial scan, measured as a linear distance between two points: the beginning of the episcleral lake anteriorly to its end posteriorly.	If the whole length of the episcleral lake could not be captured in a single image, the maximum visible length was measured.
**Maximal Cavity Height**	MCH	The maximum height of the largest cavity in blebs showing cavernous tenon changes (T3-Pattern according to JBGS) in the tangential scans, measured as the maximum perpendicular distance from the inferior to the superior edge of this cavity.	MCH was measurable only in blebs showing the T3-Pattern
**Maximum Cavity Width**	MCW	The maximum width of the largest cavity in blebs showing cavernous tenon changes (T3 pattern according to JBGS) seen in tangential scans, measured as a direct line between two points: the beginning of the cavity nasally to its end temporally.	MCW was measurable only in blebs showing the T3-Pattern
**Maximal Cavity Length**	MCL	The maximum posterior extension of the largest cavity seen in a radial scan of blebs showing cavernous tenon changes (T3 pattern), measured as a linear distance between two points: the beginning of the cavity anteriorly to its end posteriorly.	If the whole length of the cavity could not be captured in a single image, the maximum visible length was measured.
**Bleb Wall Thickness at the Lake**	BWT-L	Minimal thickness of the bleb wall at the scan with the MLH, measured as the minimal perpendicular distance between the end of the episcleral lake and the first reflex at the conjunctiva.	BWT was measurable only in blebs showing the ES1-Pattern
**Bleb Wall Thickness at the Cavity**	BWT-C	Minimal thickness of the bleb wall at the scan with the MCH, measured as the minimal perpendicular distance between the end of the cavity and the first reflex at the conjunctiva.	BWT-C was measurable only in blebs showing the T3-Pattern
**Distance to Limbus**	DtL	The linear distance between two points: point of corneal surface corresponding to the scleral spur and the point of the scleral surface corresponding to the highest point of the bleb in radial scans.	

**Table 2 diagnostics-13-02318-t002:** Patients’ characteristics and demographic data. POAG: primary open-angle glaucoma, IOP: intraocular pressure, NoM: number of medications. †: Mann–Whitney U test, ∫: chi-square test, ‡: independent-sample *t* test.

	PRESERFLO-Group (*n* = 41)	XEN-Group (*n* = 39)	*p*-Value
**Age (years)**	67.2 ± 13.0	67.4 ± 8.5	0.94 †
**Sex (male)**	18	19	0.82 ∫
**Laterality (right eye)**	22	23	0.67 ∫
**Type of Glaucoma**			0.15 ∫
**POAG** **Pseudoexfoliation** **Uveitic** **Secondary to ocular surgery**	30 2 4 5	32 3 2 2	
**Preoperative IOP (mmHg)**	22.9 ± 6.9	21.4 ± 4.6	0.26 ‡
**NoM preoperatively**	3.0 ± 1.3	2.9 ± 1.0	0.8 †
**Mean follow up (days)**	239.9 ± 244.2	300 ± 8 ± 254.8	0.24 †
**Postoperative IOP (mmHg)**	11.8 ± 3.7	13.6 ± 3.5	0.02 ‡
**NoM postoperatively**	0.2 ± 0.6	1.1 ± 2.4	0.04 †

**Table 3 diagnostics-13-02318-t003:** List of morphological changes at each level and comparison between both groups according to the JBGS. C0: no conjunctival changes, C1: conjunctival cysts, C2: subconjunctival spaces, T0: no tenon changes, T1 hyperreflective tenon changes, T2: hyporeflective tenon changes, T3: cavernous tenon changes, ES0: no episcleral lake visible, ES1: episcleral lake visible, chi-square test for all cases. Significant *p*-Values were marked in bold.

Tomographical Pattern	PRESERFLO-Group	XEN-Group	*p*-Value
**C0**	4 (9.8%)	4 (10.3%)	0.9
**C1**	5 (12.2%)	8 (20.5%)	0.9
**C2**	32 (78.0%)	27 (69.2%)	0.37
**T0**	0	1 (2.6%)	0.3
**T1**	13 (31.7%)	4 (10.3%)	**0.02**
**T2**	21 (51.2%)	20 (51.3%)	1.0
**T3**	7 (17.1%)	14 (35.9%)	**0.05**
**ES1**	38 (92.7%)	12 (30.8%)	**<0.001**

**Table 4 diagnostics-13-02318-t004:** Measurements of parameters representing bleb dimensions, all measurements in mm. ‡: independent-sample *t* test. † Mann–Whitney U. Significant *p*-Values were marked in bold.

Parameter	PRESERFLO-Group	XEN-Group	*p*-Value
**MBH**	2.13 ± 0.5	1.85 ± 0.6	**0.027** ‡
**MBW**	10.31 ± 2.3	9.1 ± 2.3	**0.021** ‡
**MBL**	9.13 ± 1.8	8.24 ± 1.9	**0.04** ‡
**MLH**	0.44 ± 0.1	0.6 ± 0.5	0.11 †
**MLW**	3.42 ± 1.4	2.8 ± 1.5	0.25 ‡
**MLL**	3.75 ± 1.5	2.47 ± 1.3	**0.014** †
**MCH**	0.7 ± 0.3	1.62 ± 2.2	0.34 †
**MCW**	2.95 ± 0.7	2.04 ± 0.9	**0.038** ‡
**MCL**	3.65 ± 1.3	2.22 ± 1.7	0.11 ‡
**BWT-L**	1.60 ± 0.5	1.1 ± 0.4	**0.004** ‡
**BWT-C**	1.84 ± 0.5	0.72 ± 0.5	**0.001** ‡
**DtL**	6.21 ± 1.2	5.21 ± 1.8	**0.005** ‡

## Data Availability

Data are available on request from the authors.

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
