# Peer review of "A Comparative Analysis of Morphology and Dimensions of Functional Blebs following PRESERFLO-Microshunt and XEN-Gel-Stent, a Study Using Anterior Segment OCT"

_diagnostics, 2023, doi:10.3390/diagnostics13142318_

Round 1

Reviewer 1 Report

Dear authors

I would like to congratulate all authors for the informative presenting an interesting quantitative measurement of the filtering blebs. The manuscript is very well written. I have only few minor comments as following.

The only disadvantage of the study was the retrospective design.

The study could not present the correlation between the bleb findings and the level of IOP, due to the study methodology (included only functioning blebs). The authors may conclude only that the bleb findings measured with AS-OCT in this study were more common in eyes underwent PRESERFLO-implantation than Xen-implantation which was probably related to the lower level of IOP.

Please carefully revise the manuscript pattern and omit unnecessary sentences such as line 186-188.

There are too many tables. Table 3 may be discarded.

In each table with comparison, please indicate the statistic comparison method, such as * t-test or ** Mann-Whitney U test.

Best regards,

Reviewer 2 Report

The article is well written, but there are a lot of bias

1) You can not compare the bleb of XEN gel stent and PRESERFLO because, as you know, the surgical approach is completely different 

2)" The decision of which procedure to perform on each patient was primarily based on the availability of the stents, with XEN being more commonly available from September 2018 to July 2020, and PRESERFLO being the predominant option thereafter. " This is a very big bias because XEN gel stent (is 45 or 65? please specify) and PRESERFLO are not comparable about the surgical indication.  Randomising the patient "on the availability" is a very big mistake because XEN and PRESER flow has different indications on surgical glaucoma treatment.  

3) On results it is not clear why  "At the tenon level, the most common pattern observed in both groups was hypo reflective changes (T2), with a prevalence of 51.2% in the PRESERFLO-Group and 51.3% in the XEN Diagnostics 2023, 13, x FOR PEER REVIEW 10 of 16 Group (p=1.0). The PRESERFLO" and in your opinion what is the clinical meaning. 

The study could be re-write only with the imaging of XEN bleb or PRESERFLO bleb during the follow up time. For example:  What are the bleb change after 6-1-year of follow up ? 

Reviewer 3 Report

Dear colleagues,

congratulations on your excellent paper. My only complaint is that the eyes that were excluded from the study due to dysfunctional blebs and the reasons for exclusion were not analyzed. I think it would be interesting to see that data.

The main issue is the difference in the morphology of the bleb in the two methods of glaucoma treatment. Elucidation of the mentioned structural changes could contribute to the decision of a more suitable and better method of glaucoma treatment in individual patients.

The topic is relevant and interesting considering the small number of works that analyze the mentioned changes, especially for a long time after surgery.

These two methods of treatment have been analyzed in many papers, but the morphological postoperative changes have not been analyzed in such detail anywhere.

The paper is well written and easy to read.

The conclusions are consistent with presented evidence.

But my suggestion written in the review still stands.
